# Resveratrol Inhibits Proliferation and Induces Autophagy by Blocking SREBP1 Expression in Oral Cancer Cells

**DOI:** 10.3390/molecules27238250

**Published:** 2022-11-26

**Authors:** Masakatsu Fukuda, Yudai Ogasawara, Hiroyasu Hayashi, Katsuyuki Inoue, Hideaki Sakashita

**Affiliations:** 1Division of Biochemistry, Department of Oral Biology and Tissue Engineering, Meikai University School of Dentistry, Saitama 350-0283, Japan; 2Division of Oral and Maxillofacial Surgery, Department of Diagnostic and Therapeutic Sciences, Meikai University School of Dentistry, Saitama 350-0283, Japan

**Keywords:** sterol regulatory element-binding protein 1, human oral squamous cell carcinoma, resveratrol, autophagy, epidermal fatty acid-binding protein

## Abstract

Resveratrol is a polyphenolic antioxidant found in grapes, red wine, and peanuts and has been reported to have anti-neoplastic effects on various cancer types. However, the exact mechanism of its anti-cancer effects in oral cancer is not fully understood and remains controversial. Resveratrol exhibits strong hypolipidemic effects; therefore, we examined its effect on lipid metabolism in oral cancer. Resveratrol significantly reduced cell viability and induced autophagic cell death in oral cancer cells but not in normal cells. This selective effect was accompanied by significantly reduced lipogenesis, which is caused by downregulation of the transcription factor sterol regulatory element-binding protein 1 (SREBP1) gene, followed by downregulation of the epidermal fatty acid-binding protein (E-FABP). It was strongly suggested that resveratrol-induced autophagy resulted from the inhibition of SREBP1-mediated cell survival signaling. Luciferase reporter assay further indicated that resveratrol has a potent and specific inhibitory effect on SREBP1-dependent transactivation. Importantly, resveratrol markedly suppressed the growth of oral cancer cells in an animal xenograft model, without exhibiting apparent cytotoxicity. In conclusion, resveratrol induces autophagy in oral cancer cells by suppressing lipid metabolism through the regulation of SREBP1 expression, which highlights a novel mechanism of the anti-cancer effect of resveratrol.

## 1. Introduction

Human oral squamous cell carcinoma (HOSCC) is the most common oral mucosa cancer. According to the GLOBOCAN 2018, the number of incident cases for lip and oral cancer in the world was estimated at 354,864, with an age-standardized incidence rate of 5.8 for men and 2.3 for women per 100,000. GLOBOCAN also estimated 177,384 deaths, with an age-standardized mortality rate of 2.8 for men and 1.2 for women per 100,000 [1]. Lip and oral cancer are rated the fourth most common malignancy occurring worldwide in men (8.7 per 100,000) [1]. It is an aggressive malignant neoplasm that is difficult to cure using conventional approaches including radio-, chemo-, and surgical therapies. Since surgical treatment often profoundly affects the quality of life and daily activities of patients with HOSCC, the use of novel therapeutic strategies is desirable along with other conventional treatments.

One other thing to note here is that it is an energy metabolism on oral cancer. Energy metabolism, a phenomenon in which cancer cells meet the fuel requirements for proliferation and invasion in a severe tumor microenvironment, is a pivotal feature of cancers [2]. In addition to alterations in glucose metabolism, commonly termed the Warburg effect, cancer cells undergo a wide range of changes in other metabolic pathways, including mitochondrial biogenesis, macromolecule biosynthesis, pentose phosphate pathway, and lipid metabolism [3,4,5,6]. Owing to their critical roles in tumor initiation, development, and metastasis, alterations in lipid metabolism, specifically fatty acid synthase (FAS), have been extensively studied in recent years [7,8]. FAS is a key enzyme involved in the synthesis of fatty acids (FAs) from acetyl-CoA, which is abundantly expressed in the liver and adipose tissue [9]. FAS has been reported to be overexpressed in several human cancers, including those of the lung, melanoma, prostate, breast, and oral cavity, and is associated with poor prognosis [10,11]. Furthermore, FAs are required for cancer cell proliferation to provide new phospholipids for cytomembranes [12]. FAs are intracellularly translocated through FA transporters, such as caveolin-1, fatty acid translocase (FAT/CD36), fatty acid transport proteins (FATP), and fatty acid-binding proteins (FABPs) [12]. FABPs can bind long-chain FAs and comprise a family of cytosolic proteins with over 10 isoforms [13]. Specifically, epidermal FABP (E-FABP) was first isolated from the epidermis, and is expressed in tissues throughout the body [14,15]. Additionally, E-FABP is associated with malignant neoplasms [16]. Studies suggest that cancer cells exhibit a unique behavior in lipid metabolism; for example, while most normal mature cells acquire FAs from the bloodstream, neoplasm cells exhibit increased de novo FA biosynthesis, most of which is modulated by sterol regulatory element-binding protein 1 (SREBP1) [17,18]. SREBP is a transcription factor and a master regulator of genes involved in regulating FA synthesis and uptake [19].

Over the last two decades, preclinical, epidemiological, and early phase clinical trials have shown a promising role of selected dietary constituents in decreasing the incidence of multiple cancers [20,21,22]. Considering the potential of such phytochemicals, it is essential to identify and develop new broad-spectrum chemopreventive agents that can be used either alone or in combination against cancer. Phytochemicals from natural sources, such as resveratrol, have recently gained interest as anticancer drugs with fewer side effects in patients with oral cancer [23].

Resveratrol (3,4′,5-trihydroxy-trans-stilbene) is a polyphenolic compound found abundantly in food and plants [23,24,25]. It has a wide spectrum of pharmacological bioactivities, including suppression of lipid metabolism as well as anti-inflammatory, anti-atherosclerotic, antioxidant, and antitumor properties [23,26]. Many preclinical studies have shown that resveratrol induces cancer cell death through apoptosis and autophagy [27,28,29], thereby exerting both preventive and therapeutic effects on cancers. Although resveratrol modulates various steps of carcinogenesis and development, the underlying mechanisms of its anti-cancer effects leading to apoptosis and autophagy are unclear in human oral cancer cells. Therefore, the present study aimed to elucidate the fundamental mechanism of the anti-cancer effects of resveratrol on oral cancer cells through modulation of lipid metabolism and, consequently, clarify a potential target for chemoprevention of oral cancer, in vitro and in vivo.

## 2. Results

### 2.1. Gene Expression Levels of E-FABP and SREBP1 in HOSCC Cell Lines

E-FABP acts as a transporter of FAs which are required for cancer cell proliferation to provide new phospholipids for cytomembranes and that SREBP is a master regulator of genes involved in regulating FA synthesis and uptake. Then, differences in E-FABP and SREBP1 gene expression levels in HOSCC cell lines (HSC-2, HSC-3, HSC-4, Ca9-22, and SAS) were firstly analyzed by qRT-PCR. The results showed that Ca9-22 cells had the highest levels of E-FABP and SREBP1 mRNA expression among the HOSCC cell lines (Figure 1).

### 2.2. Expression of E-FABP and SREBP1 Protein in HOSCC Cell Lines

SDS-solubilized extracts of HOSCC cell lines were subjected to immunoblot analysis to determine E-FABP and SREBP1 protein expression and quantity. Results revealed that all the HOSCC cell lines expressed E-FABP and SREBP1, with the highest levels in Ca9-22 cells (Figure 2). Based on these results, Ca9-22 cells derived from gingival cancer were used to further study the cellular mechanisms.

### 2.3. Impact of Resveratrol on Tumor Cell Growth

We performed a cell viability assay to examine whether resveratrol induced cell death in the Ca9-22 cells. As shown in Figure 3A, cell viability showed a significant time- and dose-dependent reduction in Ca9-22 cells upon treatment with resveratrol. Additionally, we compared the effects of resveratrol on Ca9-22 and NGE cells after 24 h of treatment with 0 and 50 µM resveratrol (Figure 3B). The results showed that Ca9-22 cells underwent morphological changes unlike NGE cells. Moreover, resveratrol treatment had a cytocidal effect on the Ca9-22 cells. To confirm whether Ca9-22 cell death resulted from apoptosis, the levels of procaspase cleavage to active caspase-8, -9, and -3/7 (markers of apoptotic activity) were analyzed. The results demonstrated decreased levels of all markers in response to resveratrol for 24 h in a dose-dependent manner, indicating that resveratrol did not cause apoptosis in Ca9-22 cells (Figure 3C). We suspected that cell death was caused by other mechanisms, such as autophagy.

### 2.4. Resveratrol-Mediated Induction of Autophagic Death in Ca9-22 Cells

To investigate the effect of resveratrol on the expression of autophagy-related markers (p62, Beclin1, and LC3) in oral cancer cells, proteins extracted from Ca9-22 cell cultures were treated with various non-cytotoxic concentrations (2–100 µM) of resveratrol and for various time periods (0, 4, 8, 12, 24, 48, and 72 h) that were quantified using immunoblot analysis. The relative levels of p62, Beclin1, and LC3-II gradually increased after treatment with resveratrol for 24 h in a dose-dependent manner (Figure 4A). A time-course analysis of these markers in Ca9-22 cells treated with 50 µM resveratrol (Figure 4B) revealed that p62, Beclin1, and LC3-II proteins were constitutively induced in Ca9-22 cells with a gradual increase in their relative quantities in a time-dependent manner, showing the highest levels at 24 h, followed by a gradual decrease up to 72 h. These findings strongly suggested that resveratrol induces autophagy in Ca9-22 cells in a dose-dependent manner.

To confirm whether resveratrol induced specifically autophagic cell death in Ca9-22 cells, an inhibition assay was performed. Ca9-22 cells were pretreated with 2 mM 3-MA [30], an autophagy inhibitor, for 1 h before exposure to 50 µM resveratrol, for 24 h. The results showed that cell viability was reduced in response to resveratrol; however, pretreatment with 3-MA inhibited resveratrol-induced cell death (Figure 4C). Moreover, treatment with C75, an FAS inhibitor, caused strong cell death in Ca9-22 cells. Additionally, p62, Beclin1, and LC3 expression levels were evaluated by immunoblot analysis. Although autophagy-related proteins were markedly increased by resveratrol treatment, 3-MA pretreatment almost completely blocked their expression in Ca9-22 cells (Figure 4D). Interestingly, increased levels of autophagy-related markers were not observed in resveratrol-treated NGE cells. Therefore, resveratrol may exert tumor-specific cytotoxic effects. These results indicated that autophagic cell death in Ca9-22 cells was indeed a result of resveratrol treatment. It should be noted that, although C75 treatment reduced FAS expression, resveratrol had no effect on FAS expression, despite inducing autophagic cell death in Ca9-22 cells. Accordingly, we focused on E-FABP as a potential therapeutic target.

### 2.5. SREBP1 Regulated E-FABP Expression in Ca9-22 Cells

To investigate the regulatory mechanism of proliferative activity in Ca9-22 cells, we analyzed SREBP1 and E-FABP expression levels in cells treated with TNF-α or resveratrol by immunoblot analysis. The results revealed that the levels of SREBP1 and E-FABP proteins were up- and down- regulated, respectively, in response to TNF-α and resveratrol, compared to those in the control cells (Figure 5A). Hence, we speculated that SREBP1, a nuclear transcription factor related to lipid metabolism, regulates E-FABP expression in Ca9-22 cells. We examined this hypothesis via inhibition of SREBP1 expression of using siRNA. SREBP1 knockdown was confirmed by immunoblot analysis (Figure 5B). Subsequently, we assessed the effect of SREBP1 knockdown on E-FABP expression and found that its protein levels were greatly reduced in SREBP1-knockdown Ca9-22 cells compared to those in control-siRNA transfected or control cells (Figure 5B). Simultaneously, the results of E-FABP knockdown demonstrated that it did not lead to downregulation of SREBP1 expression (Figure 5C) or Ca9-22 cell death (Figure 5D). Collectively, these data indicate that the downregulation of E-FABP is induced via suppression of SREBP1 activity by resveratrol. In other words, resveratrol downregulated SREBP1, which, in turn, regulated E-FABP expression and induced autophagic cell death in Ca9-22 cells.

### 2.6. Resveratrol Inhibits TNF-α-Mediated SREBP1 Activation in Ca9-22 Cells

To examine the TNF-α-mediated regulation of SREBP1 expression in Ca9-22 cells, we performed immunoblotting followed by densitometric analyses. The results revealed that SREBP1 protein initially localized in the cytoplasm of control (untreated) Ca9-22 cells (Figure 6A), then it was transported to the nucleus upon stimulation with TNF-α for 30 min (Figure 6B). However, the time-course analysis of SREBP1 protein expression up to 24 h demonstrated a gradual decrease in nuclear translocation. The regulation of SREBP1 expression in Ca9-22 cells after resveratrol treatment was also analyzed. SREBP1 was primarily localized in the cytoplasm, with small quantities detected in the cell membrane and nucleus upon stimulation with resveratrol for 30 min (Figure 6C). We examined the effect of resveratrol on TNF-α-induced nuclear translocation of SREBP1 by immunoblot analysis. SREBP1 translocation to the nucleus was detected in Ca9-22 cells stimulated with TNF-α (10 ng/mL) and resveratrol (50 µM) for 30 min (Figure 6D); however, it was markedly inhibited, compared to that in cells treated with TNF-α alone. Collectively, these results showed inhibition of the nuclear translocation of SREBP1 in resveratrol-treated Ca9-22 cells.

To investigate the effects of TNF-α and resveratrol on SREBP1-dependent transcriptional activity in Ca9-22 cells, a luciferase reporter assay was performed. The results revealed strong luciferase activity induced by TNF-α (Figure 6E). Additionally, the maximum SRE-dependent transcription was observed after treatment with 10 ng/mL of TNF-α, at 4 h (without further increase thereafter), which induced a five-fold increase in luciferase activity compared to that in cells without TNF-α. Meanwhile, the LDLR-Luc reporter construct with a single nucleotide mutation within SRE did not respond to TNF-α (Figure 6F), suggesting that the increase in luciferase activity was completely dependent on the presence of SRE sites. Moreover, luciferase activity in Ca9-22 cells decreased with time after 1 h of resveratrol (50 µM) treatment (Figure 6E), with the maximum inhibition (10-fold) at 24 h. In addition, resveratrol inhibited luciferase activity induced by TNF-α in a time-dependent manner (Figure 6E). Similarly, the level of luciferase activity was dependent on the presence of SRE sites, as mentioned above, as the LDLR-Luc reporter construct with a single nucleotide mutation within the SRE did not respond to resveratrol (Figure 6F). Collectively, these data indicated that resveratrol has a specific and potent inhibitory effect on SREBP1-dependent transactivation.

### 2.7. Suppression of Tumor Growth, Inhibition of SREBP1 and E-FABP mRNA Expression, and Induction of Autophagy in the Tumor Mass of Resveratrol-Treated Nude Mice

We examined the effects of resveratrol on tumor development in vivo, using a nude mouse model. Mice were injected subcutaneously with Ca9-22 (1 × 10^6^ cells; Figure 7A–C), and the effects of resveratrol on the extent of Ca9-22 tumor mass were evaluated after 8 weeks. The mice were euthanized, and the tumor mass was histopathologically examined. As demonstrated in Table 1, resveratrol not only prevented tumor growth, but also reduced tumor volume in a dose-dependent manner. One mouse from the control group (1/10) died unexpectedly. As shown in Figure 7D, histopathological findings revealed that the tumor mass was composed of various stratified squamous tumor cells arranged as islands with different shapes and sizes, with keratinous pearls inside. Additionally, some of the cells were acidophilic with pyknotic nuclei and karyolysis, while the rest had nuclei of different shapes and sizes, larger than the nuclei of the normal epithelium. Infiltration of inflammatory small round cells was also observed in the peritumoral stroma. After treatment with 50 µM resveratrol, the Ca9-22 tumor mass in 6 out of 10 mice was resolved. Although a tumor mass was observed in the remaining 4 mice, a clear regression of the Ca9-22 tumor mass with marked morphological changes, including autophagic vacuoles of cancer tissue (Figure 7E,F), was evident (Table 1). At the highest dose of resveratrol (100 µM per day), the Ca9-22 tumor mass was completely resolved when 1 × 10^6^ Ca9-22 cells were injected subcutaneously (none of the 10 mice showed a detectable tumor mass; Table 1 and Figure 7G). Moreover, injecting a higher number of Ca9-22 cells (1 × 10^7^) resulted in similar effects as resveratrol on the prevention of tumor growth. Next, the mRNA expression of SREBP1, E-FABP, and p62/SQSTM1 was evaluated by RNAscope ISH in the Ca9-22 tumor mass. While SREBP1 and E-FABP mRNAs were widely expressed with similar distribution patterns (Figure 8A,B), p62/SQSTM1 mRNA showed modest expression (Figure 8C) in the tumor mass without resveratrol treatment. In contrast, Ca9-22 tumor masses treated with 50 µM resveratrol showed markedly reduced expression of SREBP1 and E-FABP mRNAs (Figure 8E,F), whereas p62 mRNA was expressed only slightly (Figure 8G). To further confirm whether this reduction in cancer tissue was caused by autophagic cell death, the protein expression level of p62/SQSTM1, an autophagy-specific substrate, was examined using immunohistochemistry. We observed high levels of p62 immunoreactivity in the control mice without resveratrol treatment (Figure 8D), whereas it was almost completely absent in Ca9-22 tumor masses treated with 50 µM resveratrol (Figure 8H). These findings indicate that resveratrol effectively reduced SREBP1 and E-FABP expression and induced autophagic cell death in Ca9-22 tumor masses in nude mice.

## 3. Discussion

Resveratrol is a multifunctional polyphenol with various biological activities [28,29]. It is well documented that resveratrol can suppress the expression of genes related to lipid metabolism [31,32] and induce both autophagy and apoptosis in human cancer cells [27,28,29]. Although SREBPs are well known as master regulators of lipid metabolism [24], it is unclear whether resveratrol inhibits the action of SREBP in human oral cancer cells.

In the present study, we highlighted the SREBP1-associated effect of resveratrol on tumor growth and invasion in oral cancer using an in vitro cell culture and an in vivo nude mouse cancer model. First, our results indicate that resveratrol induced autophagic cell death, but not apoptosis, in Ca9-22 cells. Therefore, the results of our study are inconsistent with previous findings. Moreover, our results showed that the expression of FAS was not downregulated in resveratrol-treated Ca9-22 cells; therefore, we changed the target from FAS to E-FABP. In addition, resveratrol was highly selective for cancer cells. The cause of this selective cytotoxic effect of resveratrol on cancer cells, but not against NGE cells, remains elusive. Despite an extensive review of the available literature, we found no precise explanation for the mechanism underlying this selective effect exerted by resveratrol on non-cancer cells. We hypothesized that NGE cells do not undergo rigorous lipid metabolism, unlike cancer cells, and have low expression levels of SREBP1, E-FABP, and FAS; therefore, they do not serve as appropriate targets for resveratrol-mediated antioxidative activity.

According to previous studies, SREBPs serve as a key link between lipid metabolism and inflammation, energy stress, cell growth, nutrition, and other pathological and physiological processes [33]. Moreover, TNF-α stimulates SREBP1 activation via a caspase-dependent pathway in HepG2 cells derived from hepatocellular carcinoma [34]. In the present study, we used TNF-α to activate SREBP1 in Ca9-22 cells, to mimic the local inflammatory response in the invasive front of cancer, as some tumor cells were reported to produce TNF-α [35]. The results demonstrated that TNF-α-mediated activation of SREBP1 regulates E-FABP expression, as evidenced by the increased E-FABP expression in TNF-α-stimulated Ca9-22 cells and decreased E-FABP expression after SREBP1 knockdown. Additionally, it is known that nuclear translocation of the E-FABP protein mediated by lipid ligands can activate transcription factors, including SREBPs and PPARs, to initiate proliferative signaling in several cancer models [36]. Therefore, we examined SREBP1 expression in response to E-FABP knockdown in the Ca9-22 cells. These results indicated that SREBP1 expression remained unaffected, and cell death was not induced. Collectively, these findings are consistent with those of previous reports and validate our data. There are several reports available on SREBP1-mediated regulation of FAS expression [8,19]; however, we did not find sufficient reports highlighting the regulation of E-FABP expression by SREBP1. This is the first report of SREBP1-mediated regulation of E-FABP expression. We further found that TNF-α-mediated SREBP1 activation was blocked by resveratrol, suggesting that resveratrol interferes with the TNF-α signal transduction cascade at the initial step. Resveratrol treatment also inhibits the translocation of SREBP1 to the nucleus. Thus, two separate lines of evidence allowed us to conclude that resveratrol is a specific and potent inhibitor of SREBP1 activation in Ca9-22 cells: (a) resveratrol inhibited TNF-α-induced nuclear transactivation of SREBP1, and (b) resveratrol suppressed SREBP1-dependent transcription. Indeed, it has been described that SREBP1 regulates the expression of genes associated with lipid metabolism, including FASN, acetyl-CoA carboxylase 1 (ACC1), SCD1, and LDLR [37]. However, we believe that E-FABP may be an additional SREBP target. Therefore, downregulation of SREBP1 could result in the suppression of crucial effects on lipid metabolism. This study suggests that in oral cancers, resveratrol has suppressive effects of lipid metabolism and anti-cancer activities but that these results need to be confirmed using more cell lines as well as other animal models.

## 4. Materials and Methods

### 4.1. Reagents

Immunoblot analysis of E-FABP was performed using a mouse anti-human E-FABP monoclonal antibody (Mab E-FABP, #sc-365236; Santa Cruz Biotechnology, Santa Cruz, CA, USA) as the primary antibody. Rabbit anti-human p62 polyclonal antibody (Pab p62, #PM045Y), rabbit anti-human Beclin1 polyclonal antibody (Pab Beclin1, #PD017Y), and mouse anti-human LC3 monoclonal antibody (Mab LC3, #M186-3Y) were procured from Medical and Biological Laboratories (Nagoya, Japan). Rabbit anti-human sterol regulatory element-binding protein 1 monoclonal antibody (Mab SREBP1, #NB600-582) was procured from Novus Biologicals (Centennial, CO, USA). Mouse anti-human fatty acid synthase (FAS) monoclonal antibody (Mab FAS, #10038) was procured from Immuno-Biological Laboratories Co., Ltd. (Fujioka, Japan). The rabbit anti-human β-actin monoclonal antibody (Mab β-actin, #4970) was obtained from Cell Signaling Technology (Tokyo, Japan). Resveratrol (3, 4′, 5-trihydroxy-trans-stilbene) and 3-methyladenine (3-MA; PI3K class III inhibitor) were procured from Sigma-Aldrich (St. Louis, MO, USA). Recombinant human TNF-α (R&D Systems, Inc., Minneapolis, MN, USA) was used to stimulate the Ca9-22 cells. For in situ hybridization, probes against human SREBP1 (ACD# 469871), human p62/SQSTM1 (ACD# 415881), and human E-FABP (ACD# 566111-C3) were procured from Advanced Cell Diagnostics Inc. (ACD; Newark, CA, USA).

### 4.2. Cell Culture

HOSCC cell lines (HSC-2, HSC-3, HSC-4, Ca9-22, and SAS; Japanese Cancer Research Resources Bank, Osaka, Japan) mycoplasma testing has been carried out, and cell lines were cultured independently in 25 cm^2^ culture flasks with RPMI-1640 medium containing 10% heat-inactivated fetal bovine serum and 1% antibiotic-antimycotic (Life Technologies, Tokyo, Japan). Cells were grown to confluency, at 37 °C, in an atmosphere of 5% CO_2_. Normal human gingival progenitor (NHGP) cells (CELLnTEC advanced cell systems, Bern, Switzerland) were maintained in gingival progenitor cell maintenance medium (CELLnTEC advanced cell systems) without antibiotics on 10 cm^2^ polyethyleneimine-coated glass plates until differentiation into normal gingival epithelial (NGE) cells.

### 4.3. RNA Extraction and Quantitative Reverse Transcription-Polymerase Chain Reaction (qRT-PCR)

Total RNA was extracted from monolayered HOSCC cells (1 × 10^6^ cells/mL) using the AGPC method, as described previously [38]. The expression patterns of E-FABP and SREBP1 were confirmed by qRT-PCR analyses using a Bio-Rad iCycler system (Bio-Rad, Tokyo, Japan) and the iScript One-Step RT-PCR kit with SYBR Green I (Bio-Rad), according to the manufacturer’s instructions and a previously described method [38]. The PCR primers were designed and synthesized by Sigma-Aldrich, Inc. (Ishikari, Japan) with the following primer sequences:

E-FABP: forward, 5′-GCC GCC GTT ATA AAG CAG CC-3′; E-FABP reverse, 5′-GCA AAG CTA TTC CCA CTC CTA GC-3′; SREBP1 forward, 5′-AAT CTG GGT TTT GTG TCT TC-3′; SREBP1 reverse, 5′-AAA AGT TGT GTA CCT TGT GG-3′; GAPDH forward, 5′-CAG CCT CAA GAT CAT CAG CA-3′; and GAPDH reverse, 5′-ACA GTC TTC TGG GTG GCA GT-3′. GAPDH mRNA with previously described primer sequences was used (set at 1) as an internal control [38].

### 4.4. Immunoblot Analysis

Proteins were extracted from monolayered HOSCC cells (1 × 10^6^ cells/mL) and quantified as previously described [38]. Immunoblot analysis was performed as previously described [38]. MAb E-FABP (1:1000), PAb SREBP1 (1:1000), PAb p62 (1:1000), PAb Beclin1 (1:1000), MAb LC3 (1:1000), and MAb FAS (1:1000) were used as primary antibodies. Horseradish peroxidase (HRP)-labeled goat anti-rabbit IgG (H + L) or anti-mouse IgG (H + L) antibody (1:25,000) (GE Healthcare, Piscataway, NJ, USA) was used as the secondary antibody. MAb β-actin (1:5000) was used as the internal control. Ca9-22 cells were grown as monolayer culture, and treated with 10 ng/mL TNF-α and/or 50 µM resveratrol for 24 h followed by immunoblot analysis for the detection of SREBP1 and E-FABP.

### 4.5. Cell Viability Assay

The assay was performed as previously described method [38]. Briefly, Ca9-22 cells (2 × 10^4^ cells/100 μL/well) were plated in a 96-microwell plate. Subsequently, the cells were treated with or without 2 nM, 50 nM, 100 nM, 1 µM, 50 µM, and 100 µM resveratrol for 24 h. WST-8/ECS solution (10 μL; Dojindo Laboratories, Tokyo, Japan) was then added to each well and incubated, at 37 °C, in a 5% CO_2_ incubator for 4 h. The cells were oscillated for 1 min, and absorbance was measured at 450 nm using a microplate reader (Bio-Rad). Next, inhibition assay against autophagy was performed. Ca9-22 cells were individually pretreated with 2 mM 3-MA (an autophagy inhibitor) for 1 h, followed by treatment with or without 50 μM resveratrol for 24 h. Thereafter, cell viability assay and immunoblot analysis were performed as described above. Cellular autophagic vacuole morphology was examined and photographed using a phase-contrast microscope (Olympus, Tokyo, Japan).

### 4.6. Apoptosis Detection Assay

The assay was performed as previously described method [38]. Briefly, Ca9-22 cells (5 × 10^3^ cells/100 μL/well) were plated in white-walled 96-well tissue culture plates and incubated for 24 h. The cells were treated with or without 2 nM, 50 nM, 100 nM, 1 µM, 50 µM, and 100 µM resveratrol for 24 h. Thereafter, the activities of caspase-3/7, -8, and -9 were determined using Caspase-Glo^®^ 3/7, 8, and 9 assays (Promega, Madison, WI, USA) according to the manufacturer’s instructions. Subsequently, 50 μL of Caspase-Glo reagent was added to each well and incubated for 30 min, followed by the measurement of luminescence as relative light units (RLUs) using a Veritas Microplate Luminometer (Promega).

### 4.7. RNA-Mediated Interference

Small interfering RNAs (siRNAs) specific for human E-FABP, SREBP1, and scrambled (control) siRNAs were synthesized by Sigma-Aldrich (St. Louis, MO, USA). The sense and antisense strand sequences of the oligonucleotides were as follows: E-FABP siRNA sense, UGU ACC CUG GGA GAG AAG U; antisense, ACU UCU CUC CCA GGG UAC A; control siRNA sense, GAU CAU GAG CGG UGC GUA A; antisense, UUA CGC ACC GCU CAU GAU C; SREBP1 siRNA sense, GAG GCA AGA CCG AAG UAA A; antisense, UUU ACU UCG GUC UUG CCU C; and control siRNA, MISSION^®^ siRNA universal negative control. Before transfection, FuGENE HD transfection reagent (Promega) was mixed with 100 nM E-FABP and 100 nM SREBP1 or 100 nM control siRNA (3:3.4 μL) in serum-free medium to a total volume of 500 μL followed by incubation for 30 min, at room temperature (RT). For SREBP1 or E-FABP knockdown, Ca9-22 cells (derived from gingival cancer; 1 × 10^5^ cells/mL) were seeded on 24-well plates and rinsed with serum-free medium, and transfected with either an SREBP1, an E-FABP siRNA duplex, or a control siRNA using FuGENE HD transfection reagents for 48 h, at 37 °C. The cells were then subjected to cell viability assay, apoptosis detection assay, and immunoblot analysis.

### 4.8. Analysis of SREBP1 Translocation

To examine SREBP1 translocation to the nucleus, we used a subcellular proteome extraction kit (S-PEK; Calbiochem, Darmstadt, Germany) according to the manufacturer’s instructions to extract cytoplasm, cell membrane and nucleus fractions of Ca9-22 cells. Ca9-22 cells were treated with 10 ng/mL of TNF-α and/or 50 µM resveratrol. Then, the assay was subsequently performed as previously described method [39]. Thereafter, each sample was subjected to immunoblot analysis followed by densitometric analyses. Filters were scanned and computer-generated images were analyzed with the ImageJ program to obtain densitometric values. For each series of samples (cytoplasm, cell membrane and nucleus), the relative density of each image was calculated and expressed as a percentage of the value (arbitrarily set at 100) indicated by a sharp sign.

### 4.9. Plasmids

The pGL4 firefly luc2 plasmid and hRluc-TK-renilla were procured from Promega. The low-density lipoprotein receptor plasmid (pLDLR)-Luc construct (also known as pES7, Addgene plasmid #14940), harboring the SREBP-responsive Sterol Responsive Element (SRE) sequence (ATCACCCCAC), and the pLDLR-Luc mutSRE construct (LDLR-Luc MUT, Addgene plasmid #14945), harboring an SREBP-unresponsive mutant SRE (ATAACCCCAC) were obtained from Addgene (Watertown, MA, USA). A PCR fragment containing nucleotides 335–3 of the human LDL receptor gene was cloned into the SmaI site of pGL2-basic to generate pLDLR-Luc.

### 4.10. Transfection of Ca9-22 Cells with Plasmids and Luciferase Reporter Assay

Ca9-22 cells (1 × 10^5^ cells/mL) were cultured for 12 h in 24-well culture plates containing RPMI1640 medium supplemented with 10% FBS. Cells were transiently transfected with pLDLR-Luc, pLDLR-Luc mutSRE, and pGL4-basic plasmids (firefly) as control and hRluc-TK reference Renilla luciferase plasmids (Promega) using FuGENE HD transfection reagents (Roche, Nutley, NJ, USA), according to the manufacturer’s instructions. After 24 h of transfection, cells were treated with 10 ng/mL TNF-α and/or 50 µM resveratrol for 24 h. Subsequently, cells were lysed, and firefly and Renilla luciferase activities were analyzed using the Dual-Luciferase System (Promega) according to the manufacturer’s instructions. To standardize transfection efficiencies, luciferase activity from pLDLR-Luc and pLDLR-Luc mutSRE was normalized to Renilla luciferase activity.

### 4.11. Nude Mouse Model of Ca9-22 Tumor Mass

All experimental procedures were performed with approval from the Animal Experimentation Committee of our university. Specific pathogen-free athymic four-week-old BALB/c female mice were kept under sterile conditions in a laminar flow room in cages with filter bonnets and fed a sterilized mouse diet and water. Mice were anesthetized with Isoflurane. Mice were subcutaneously injected with Ca9-22 cells (1 × 10^6^ cells in 100 μL of PBS) into the back using a 27-gauge needle. Tumor size was measured using calipers daily in all mice, and movable and elastic-hard Ca9-22 tumor masses grew to approximately 10 mm in diameter after three weeks of Ca9-22 cell inoculation. To examine the effects of polyphenols, the mice were treated with 50 and 100 µM/day of resveratrol and PBS (control), respectively, by intratumoral injection for seven consecutive days after the Ca9-22 tumor mass had reached approximately 10 mm in diameter. Subsequently, resveratrol or PBS (control) was administered once a week for an additional five weeks (a total of eight weeks after Ca9-22 cell implantation).

### 4.12. Quantification of Tumor Mass in Nude Mice

After eight weeks of Ca9-22 cell implantation, the mice were euthanized, and the tumor mass status was evaluated quantitatively (Table 1). Animals were transcardially perfused with 200 mL of 0.9% saline containing heparin (10,000 U/l), followed by 200 mL of phosphate-buffered 4% paraformaldehyde. The tumor volume (V, mm^3^) was calculated as 0.5 × L × W^2^, where L and W refer to the length and width (in mm), respectively. The percentage of tumor growth inhibition is expressed as the mean value (±SD) of tumor volumes (calculated for all groups with 10 mice each) relative to the volume of tumors injected with control PBS. The back skin was excised and the tumor tissue was post-fixed in 4% paraformaldehyde for 18–24 h. Post-fixation, the tumor was equilibrated to 15% and 30% sucrose and then cut into 10–15 μm-thick sections on a freezing, sliding stage microtome. Sections were stored at −80 °C until processed for RNAscope in situ hybridization (ISH) and immunohistochemistry (IHC).

### 4.13. RNAscope ISH

Detection of SREBP1, p62/SQSTM1, and E-FABP mRNAs by RNA ISH was performed on frozen slices using the RNAscope 2.5 HD Reagent Kit-Brown (ACD) according to the manufacturer’s instructions. The positive and negative control probes were used in this study. Briefly, frozen slides were dried in an oven, at 60 °C, for 30 min prior to incubation in cold 4% PFA for 15 min. The slides were then dehydrated using 50%, 70%, and 100% ethanol for 5 min each at RT, followed by H_2_O_2_ addition and incubation for 10 min at RT. For antigen retrieval, the sections were boiled (98–102 °C) in a target retrieval solution (ACD) for 5 min. After washing the slides with pure H_2_O twice for 30 s each, at RT, they were dehydrated using 100% ethanol for 3 min and air-dried for 5 min, at RT. The slides were then baked, at 60 °C, for 30 min, and a hydrophobic barrier was formed around the tissue using ImmEdge Hydrophobic Barrier Pen (ACD), followed by protein digestion using protease III treatment for 30 min, at 40 °C. Frozen sections were washed twice with pure H_2_O for 1 min each. Next, the target probes were added and allowed to hybridize at, 40 °C, for 2 h. The detection kit was used as follows: amplification steps 1–4 were performed, at 40 °C (AMP1 30 min, AMP2 15 min, AMP3 30 min, and AMP4 15 min), followed by steps 5–6, at RT (AMP5 30 min and AMP6 15 min). Sections were then incubated with diaminobenzidine (DAB) for 10 min, counterstained with Gill’s hematoxylin for 30 s, and incubated for 2 min, at RT. Washing steps between the addition of reagents were performed on an automated platform, at RT. Finally, the slides were removed, immersed in deionized water, dehydrated, cleared, and mounted. Brown punctate signals colocalized with nuclei and/or the cytoplasm of tumor cells were designated as positive.

### 4.14. Immunohistochemistry

p62 immunostaining was performed as described previously [38]. Diluted MAb p62 (1:1000) and goat anti-rabbit IgG (H + L) antibodies for p62 (1:200) were used as the primary and secondary antibodies, respectively.

### 4.15. Statistical Analysis

Results were compared between different groups using two-tailed Student’s *t*-test. Differences were considered statistically significant at *p*-values < 0.05. All analyses were performed using StatView statistical software (version 5.0; SAS Institute Inc., Cary, NC, USA). Each column and error bar represent the mean values ± SD of three independent experiments (*n* = 3 experiments; mean values ± SD).

## 5. Conclusions

Resveratrol inhibits the transactivation of SREBP1 with subsequent downregulation of E-FABP expression. Resveratrol blocked Ca9-22 cell proliferation, ultimately inducing autophagic cell death and preventing the growth of Ca9-22 tumor masses in nude mice. This molecular cascade may provide the mechanism by which resveratrol suppresses the development of oral squamous cell carcinomas. Our data also suggest that the anti-cancer activity of resveratrol may rely on the inhibition of SREBP1. However, the mechanism by which resveratrol interferes with SREBP1 activation remains unclear. Further investigation into the role of SREBP1 will help unfold lipid metabolism-mediated cancer proliferation and establish a resveratrol-based therapeutic strategy for oral cancer.

## Figures and Tables

**Figure 1 molecules-27-08250-f001:**
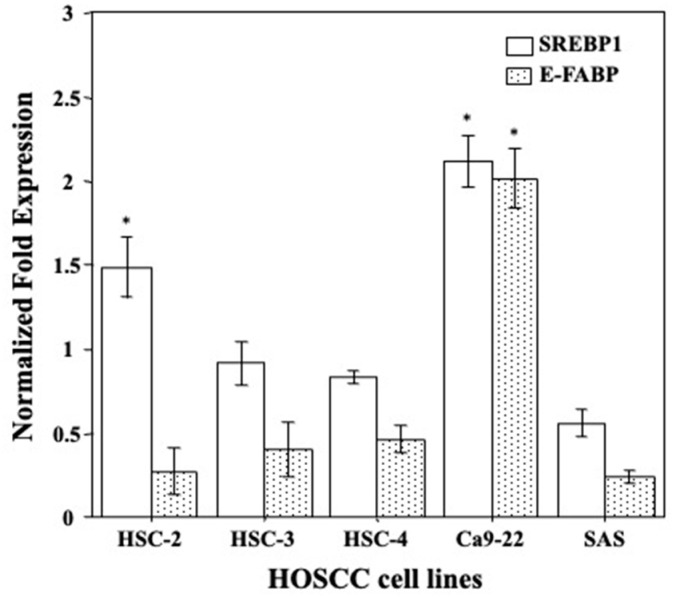
Differences in the levels of SREBP1 and E-FABP mRNA expression in HOSCC cell lines. Ca9-22 cells had the highest levels of SREBP1 and E-FABP mRNA expression. Each column and error bar represent the mean values ± SD of three independent experiments. All *p*-values: * *p* < 0.05 compared to relative internal control.

**Figure 2 molecules-27-08250-f002:**
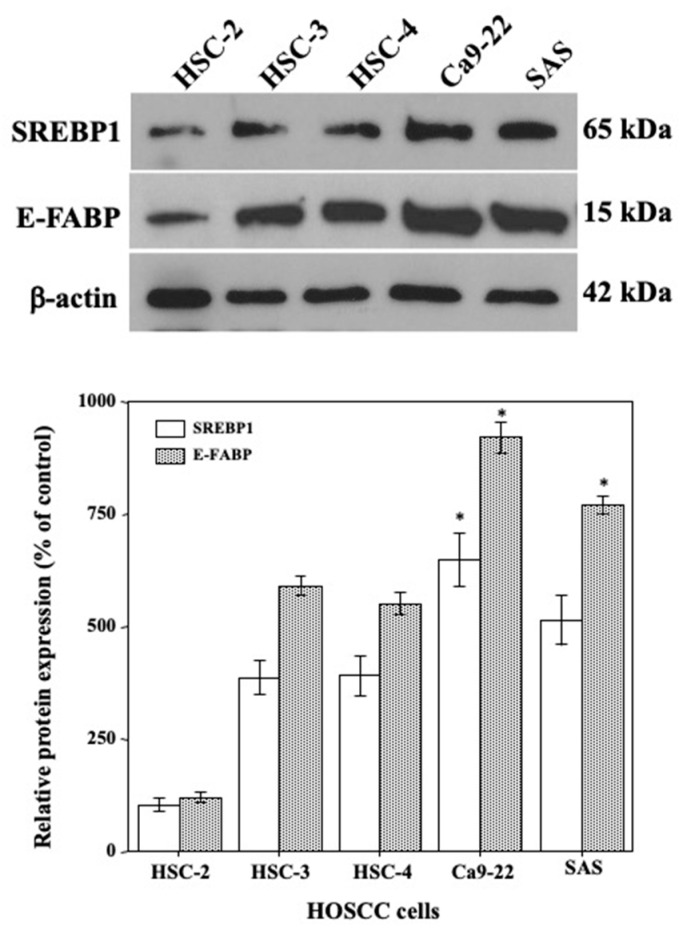
Expression profiles of E-FABP and SREBP1 proteins in HOSCC cell lines. SREBP1 and E-FABP are expressed in all HOSCC cells as 65 and 15 kDa peptides, respectively, with the highest expression levels in Ca9-22 cells. Blots are representative of *n* = 3 biological replicates. *p*-values: * *p* < 0.05 compared to β-actin control.

**Figure 3 molecules-27-08250-f003:**
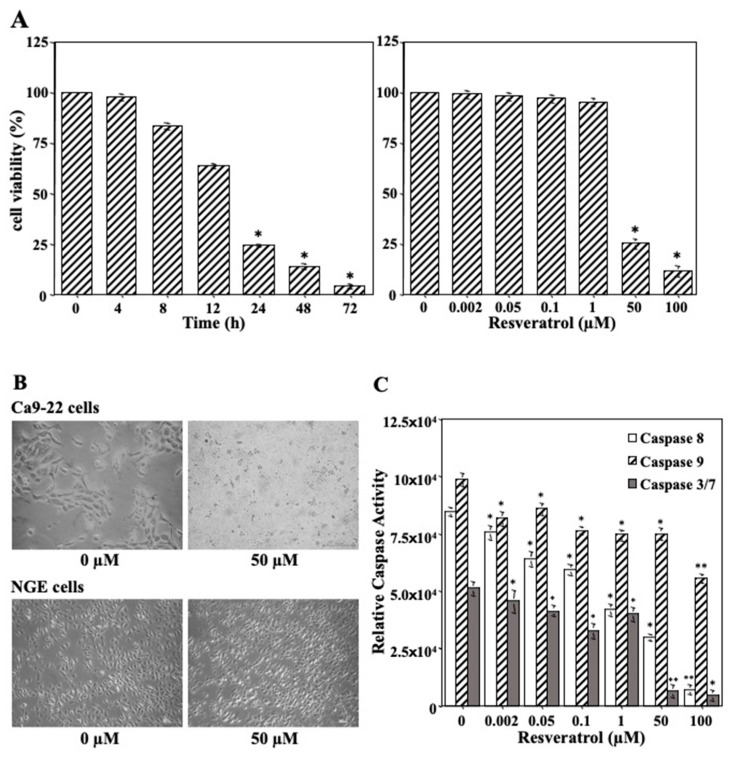
Effect of resveratrol on cell viability, morphological characteristics, and apoptotic cell death in Ca9-22 cell line. (**A**) Cell viability showed a significant time- and dose-dependent reduction in Ca9-22 cells upon treatment with 50 µM of resveratrol for various hours and with various concentrations of resveratrol for 24 h. Each column and error bar represent the mean values ± SD of three independent experiments. * *p* < 0.05 compared to untreated control. (**B**) Ca9-22 cells treated with 50 µM of resveratrol for 24 h and observed for the morphological changes. (**C**) Levels of all markers were decreased in response to resveratrol for 24 h in a dose-dependent manner, indicating that resveratrol did not cause apoptosis in Ca9-22 cells. Each column and error bar represent the mean values ± SD of three independent experiments. All *p*-values: * *p* < 0.05, ** *p* < 0.01 compared to untreated control. NGE cells: normal gingival epithelial cells.

**Figure 4 molecules-27-08250-f004:**
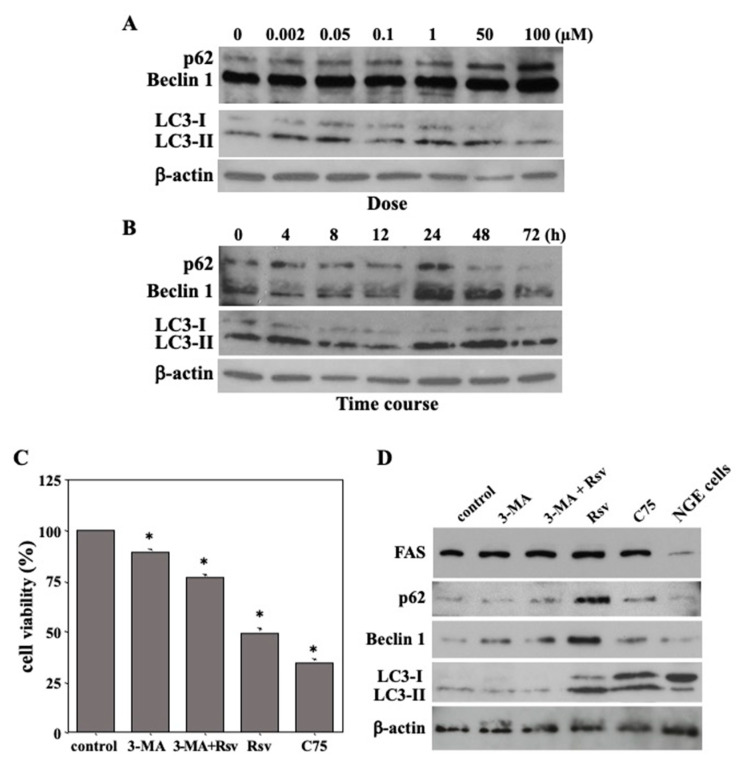
Resveratrol-induced autophagic cell death in Ca9-22 cells. (**A**) Relative levels of p62, Beclin1, and LC3-II gradually increased in a dose-dependent manner after treatment with various concentrations (0, 2 nM, 50 nM, 100 nM, 1 µM, 50 µM, and 100 µM) of resveratrol for 24 h. (**B**) Relative levels of p62, Beclin1, and LC3-II proteins were gradually increased after treatment with 50 µM of resveratrol for various time periods (0, 4, 8, 12, 24, 48, and 72 h) in Ca9-22 cells in a time-dependent manner, showing the highest levels at 24 h, followed by a gradual decrease up to 72 h. (**C**) Cells were treated with 50 µM of resveratrol, 2 mM of 3-MA or C75 for 24 h. Cell viability was reduced in response to resveratrol; however, pre-treatment with 3-MA inhibited resveratrol-induced cell death. Each column and error bar represent the mean values ± SD of three independent experiments. * *p* < 0.05 compared to untreated control. (**D**) Although autophagy-related proteins were markedly increased by resveratrol treatment, 3-MA pretreatment almost completely blocked their expression in Ca9-22 cells. (**E**) The band intensities of (**A**) are graphically represented as the relative expressions of p62, Beclin1, and LC3-I,II/β-actin. (**F**) The band intensities of (**B**) are graphically represented as the relative expressions of p62, Beclin1, and LC3-I,II/β-actin. (**G**) The band intensities of (**D**) are graphically represented as the relative expressions of p62, Beclin1, and LC3-I,II/β-actin. Immunoblots are representative of *n* = 3 biological replicates. All *p*-values: * *p* < 0.05 compared to β-actin control. 3-MA: 3-Methyladenine; Rsv: resveratrol; C75: fatty acid synthase (FAS) inhibitor; NGE cells: normal gingival epithelial cells.

**Figure 5 molecules-27-08250-f005:**
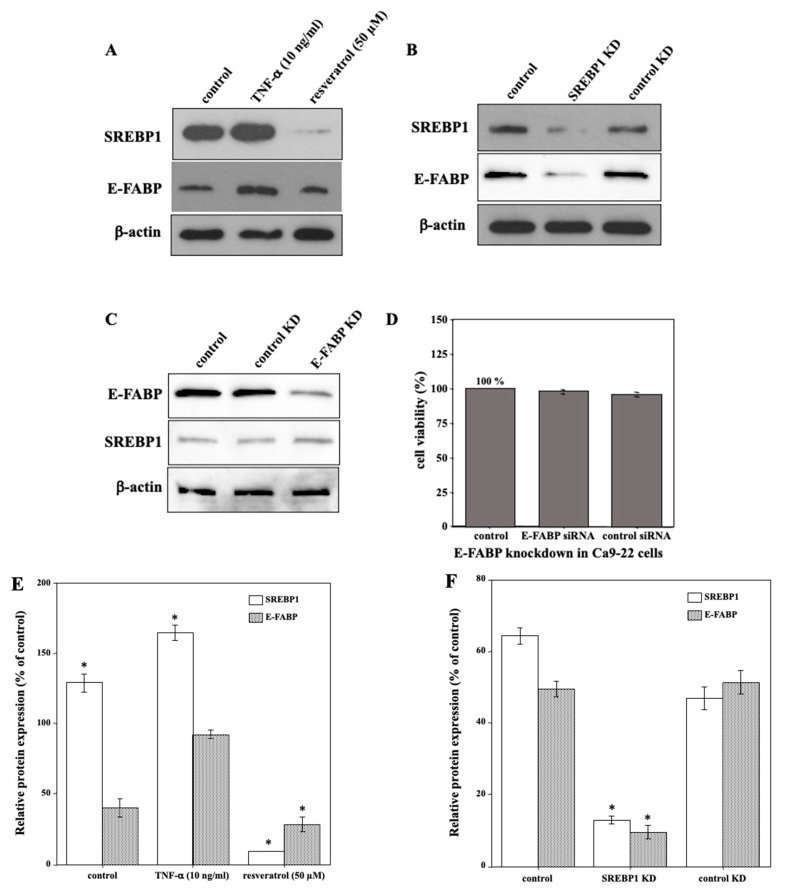
SREBP1 regulates E-FABP expression in Ca9-22 cells. (**A**) The levels of SREBP1 and E-FABP proteins were upregulated in response to 10 ng/mL TNF-α, and they were downregulated in response to 50 µM resveratrol for 24 h, respectively, compared to those in the control cells. (**B**) SREBP1 and E-FABP protein levels were markedly reduced by siRNA in Ca9-22 cells compared to those in control siRNA-transfected or control cells. (**C**) E-FABP knockdown did not lead to downregulation of SREBP1 expression. (**D**) E-FABP knockdown did not induce Ca9-22 cell death either. (**E**) The band intensities of (**A**) are graphically represented as the relative expressions of SREBP1, and E-FABP/β-actin. (**F**) The band intensities of (**B**) are graphically represented as the relative expressions of SREBP1, and E-FABP/β-actin. (**G**) The band intensities of (**C**) are graphically represented as the relative expressions of E-FABP, and SREBP1/β-actin. Each column and error bar represent the mean values ± SD of three independent experiments. Immunoblots are representative of *n* = 3 biological replicates. All *p*-values: * *p* < 0.05 compared to β-actin control.

**Figure 6 molecules-27-08250-f006:**
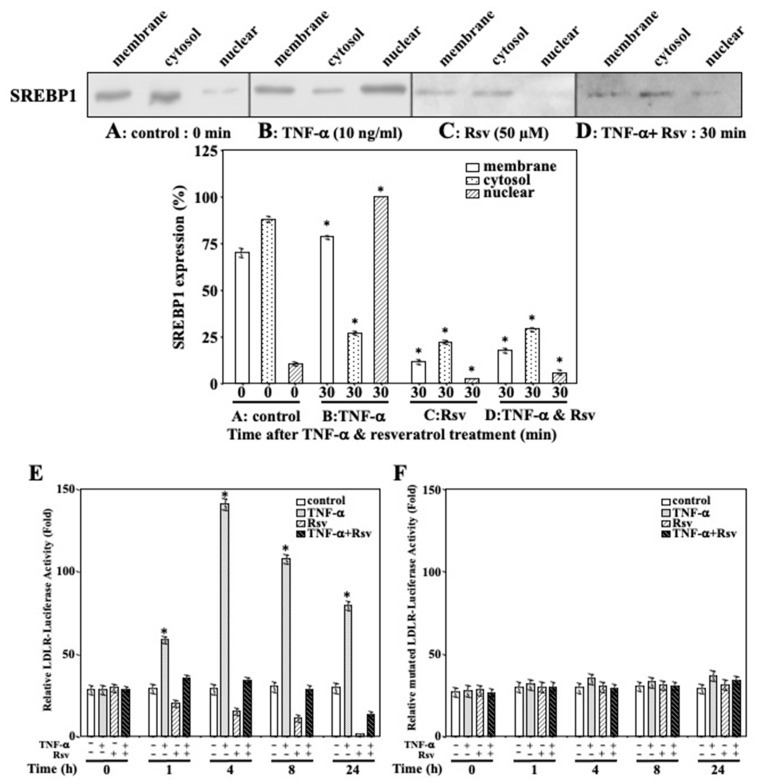
Effect of resveratrol on TNF-α-mediated SREBP1 activation in Ca9-22 cells. SREBP1 protein expression in the different cellular fractions was tested with immunoblotting in control (untreated) Ca9-22 cells (**A**), cells treated with 10 ng/mL TNF-α for 30 min (**B**), Ca9-22 cells stimulated with 50 µM resveratrol for 30 min (**C**), and Ca9-22 cells stimulated with TNF-α (10 ng/mL) and resveratrol (50 µM) for 30 min (**D**). Immunoblots are representative of *n* = 3 biological replicates. Each column and error bar represent the mean values ± SD of three independent experiments. * *p* < 0.05 compared to untreated control. (**E**) Maximum SRE-dependent transcription was observed with 10 ng/mL of TNF-α at 4 h, which induced a five-fold increase in luciferase activity compared with that in cells without TNF-α. Moreover, luciferase activity in Ca9-22 cells decreased with time after one hour of resveratrol (50 µM) treatment, with maximum inhibition (10-fold) at 24 h. Additionally, resveratrol inhibited luciferase activity induced by TNF-α in a time-dependent manner. (**F**) the LDLR-Luc reporter construct with a single nucleotide mutation within SRE did not respond to TNF-α and/or resveratrol. Each column and error bar represent the mean values ± SD of three independent experiments. All *p*-values: * *p* < 0.05 compared to untreated control.

**Figure 7 molecules-27-08250-f007:**
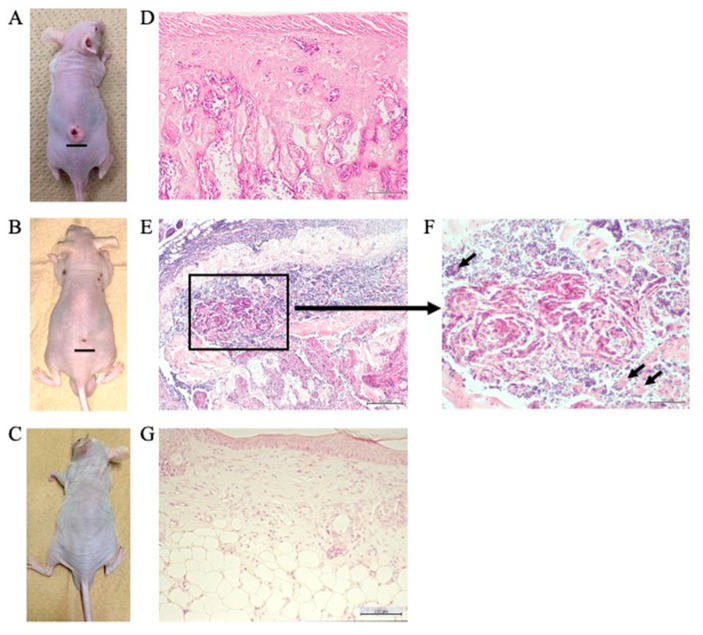
Suppression of growth in the Ca9-22 tumor mass of resveratrol-treated nude mice. Tumor appearance in mice after eight weeks of Ca9-22 cell inoculation, and five weeks of saline (control) administration (**A**), 50 µM of resveratrol (**B**) or 100 µM of resveratrol (**C**) administration. Scale bar = 10 mm. (**D**) Histopathological findings revealed that the tumor mass was composed of various stratified squamous tumor cells, arranged as islands with different shapes and sizes, with keratinous pearls inside (Hematoxylin and eosin (H-E) stain; scale bar = 200 µm, original magnification ×40). (**E**) Treatment with 50 µM of resveratrol resulted in marked morphological changes in which Ca9-22 tumor masses had formed (H-E stain; scale bar = 200 µm, original magnification ×40). (**F**) Marked morphological changes were observed, including formation of autophagic vacuoles in Ca9-22 tumor mass indicated by arrows (H-E stain; scale bar, 200 µm, original magnification ×100). (**G**) At the highest dose of resveratrol (100 µM/day), the Ca9-22 tumor mass is completely resolved when 1 × 10^6^ Ca9-22 cells were injected subcutaneously (H-E stain; scale bar = 200 µm, original magnification ×40).

**Figure 8 molecules-27-08250-f008:**
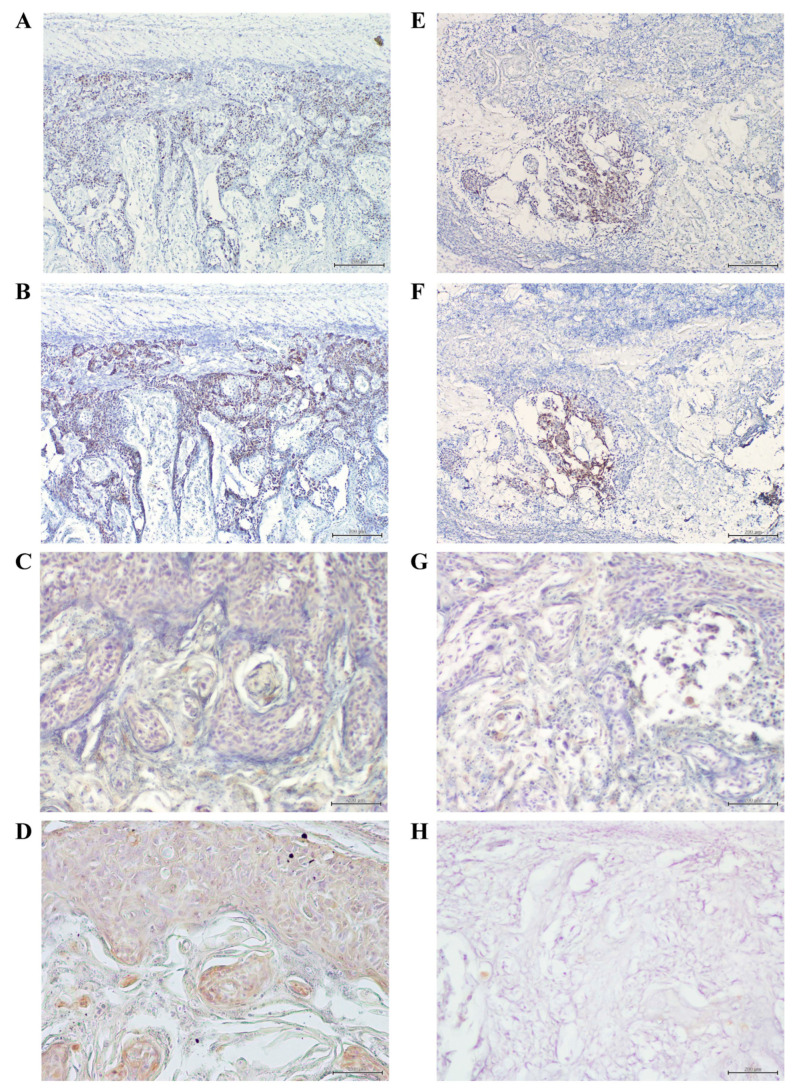
Inhibition of SREBP1 and E-FABP mRNA expression and induction of autophagy in the Ca9-22 tumor mass of resveratrol-treated nude mice. Expressions of both SREBP1 (**A**) and E-FABP (**B**) mRNAs were extensively detected with higher levels in Ca9-22 tumor mass by RNAscope ISH (Scale bar = 200 µm, original magnification ×40) in control (not exposed to resveratrol). In Ca9-22 tumor treated with 50 µM of resveratrol, expressions of both SREBP1 (**E**) and E-FABP (**F**) mRNAs were extremely restricted (Scale bar = 200 µm, original magnification ×40). ISH revealed slight expression of p62 mRNA in resveratrol-treated tumor (**G**), compared to that in the control (**C**) (Scale bar = 200 µm, original magnification ×40). Immunohistochemistry revealed high levels of p62 immunoreactivity in untreated control (**D**), compared with that in Ca9-22 tumor mass treated with 50 µM of resveratrol (**H**) (Scale bar = 200 µm, original magnification ×40). Image analyses of SREBP1, E-FABP and p62 expression in resveratrol-treated Ca9-22 tumor mass was determined using ImageJ program. All *p*-values: * *p* < 0.05 compared to untreated control.

**Table 1 molecules-27-08250-t001:** Effect of resveratrol at 8 weeks after subcutaneous inoculation of Ca9-22 cells (1 × 10^6^ cells) in nude mice.

		Control (Saline)	Resveratrol (µM/Day)
			50 100
Animals with tumor formation, n (%)		9/10 (90%)	4/10 (40%) * 0/10 (0%) **
Tumor volume ^#^, mm^3^		608.9 ± 11.82 (100%)	55.28 ± 2.87 (9.08%) * 0 (0%) **

A mouse in the control group (1/10) died unexpectedly. ^#^ Tumor volume was calculated as width^2^ × length × 0.5 and values represent mean ± SD (% relative to control). ** p* < 0.05, *** p* < 0.001 (compared to the control).

## Data Availability

The data presented in this study are available on request from the corresponding author.

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
