# Peer review of "Resveratrol Inhibits Proliferation and Induces Autophagy by Blocking SREBP1 Expression in Oral Cancer Cells"

_molecules, 2022, doi:10.3390/molecules27238250_

Round 1
Reviewer 1 Report
The authors presented their work on natural product resveratrol in the present study. They found resveratrol could induce autophagic cell death in oral cancer cells and reduce lipogenesis through SREBP1 and E-FABP. The finding is exciting and provides evidence for a novel potential therapeutic strategy in oral cancer.
The biggest concern is the concentration of resveratrol used in all the experiments. As shown in Fig 3, 50 uM of resveratrol dramatically diminished Ca9-22 cells. I don't think further cellular assays using resveratrol at 50 uM could provide supportive data since there are few cells.
Besides, they proposed a mechanism that resveratrol could down-regulate SREBP1 and thus reduce the transcription of E-FABP with very little supportive evidence. I think the hypothesis still needs more data to prove it.
Author Response
Dear Reviewer,
Thank you for your comments.
However, I can not revise in this situation.
Could you give me any more concrete comments?
Or do you mean you would like to reject this manuscript?
Reviewer 2 Report
In the present article titled “Resveratrol Inhibits Proliferation and Induces Autophagy by Blocking SREBP1 Expression in Oral Cancer Cells”, the authors have shown that Resveratrol induces autophagy-dependent cell death in oral cancer cells. There are some observations that need to be addressed.
1. In the abstract, reframe the sentence with clear meaning, “Activation of the autophagic pathway in oral cancer cells was suppressed by 3-methyladenine, suggesting that resveratrol-induced autophagy resulted from the inhibition of SREBP1-mediated cell survival signaling.”
2. The introduction is well-written, but it needs to start with information about cancer and mortality rates. Also, check for fluency and sentence connectivity.
3. In the method section, incorporate catalog number for antibodies, that will help to make reproducibility of data easily.
4. Reframe the sentence ”Confluent cell cultures were grown at 37 °C in an atmosphere of 5% CO2”…..like cells were grown till confluence.
5. Correctly annotate primer sequences like 3’-5’ directions.
6. Statistical analysis in missing in fig. 1.
7. In figure 3, how cell viability graph have the same single * significance in a different time and dose of resveratrol.
8. In microscopy images, scale bar is missing.
9. In the results section, you have misinterpreted the p62 data….in the case of functional autophagy, p62 acts as a substrate and in turn, gets degraded, whereas, in autophagy block, p62 accumulates. You have shown that resveratrol induces p62 accumulation. Kindly recheck it. Also show autophagy flux analysis in the presence of autophagy inhibitor Baf-A1.
10. Quantify all western blots that make readability and analysis easier.
Author Response
Thank you for your advices to my manuscript.
Q1. In the abstract, reframe the sentence with clear meaning, “Activation of the autophagic pathway in oral cancer cells was suppressed by 3-methyladenine, suggesting that resveratrol-induced autophagy resulted from the inhibition of SREBP1-mediated cell survival signaling.”
A1. I have changed this sentence “Activation of the autophagic pathway in oral cancer cells was suppressed by 3-methyladenine, suggesting that resveratrol-induced autophagy resulted from the inhibition of SREBP1-mediated cell survival signaling.” to “It was strongly suggested that resveratrol-induced autophagy resulted from the inhibition of SREBP1-mediated cell survival signaling.”
Q2. The introduction is well-written, but it needs to start with information about cancer and mortality rates. Also, check for fluency and sentence connectivity.
A2. I have added this sentence in the beginning of Introduction “According to the GLOBOCAN 2018, the number of incident cases for lip and oral cavity cancer in the world was estimated at 354,864, with an age-standardized incidence rate, 5.8 for men and 2.3 for women per 100,000. GLOBOCAN also estimated 177,384 deaths, with an age standardized mortality rate, 2.8 for male and 1.2 for female per 100,000 [1]. Lip and oral cavity cancer are rated the fourth most common malignancy occurring worldwide in men (8.7 per 100,000) [1].” and reframed the Introduction.
Q3. In the method section, incorporate catalog number for antibodies, that will help to make reproducibility of data easily.
A3. In the Materials and Methods section, I have added the catalog number for all antibodies which we used in this study.
Q4. Reframe the sentence ”Confluent cell cultures were grown at 37 °C in an atmosphere of 5% CO2”…..like cells were grown till confluence.
A4. I have changed this sentence “Confluent cell cultures were grown at 37 °C in an atmosphere of 5% CO2” to “Cells were grown to confluency at 37 °C in an atmosphere of 5% CO2.”
Q5. Correctly annotate primer sequences like 3’-5’ directions.
A5. I have done them.
Q6. Statistical analysis in missing in fig. 1.
A6. I have added statistical analysis in Fig. 1.
Q7. In figure 3, how cell viability graph have the same single * significance in a different time and dose of resveratrol.
A7. Asterisk (*) indicates statistically significant difference (p<0.05) from control cells.
I recalculated p-value and revised modestly the graphs.
Q8. In microscopy images, scale bar is missing.
A8. I have missed scale bar only in Fig. 7G, then I have added.
Q9. In the results section, you have misinterpreted the p62 data….in the case of functional autophagy, p62 acts as a substrate and in turn, gets degraded, whereas, in autophagy block, p62 accumulates. You have shown that resveratrol induces p62 accumulation. Kindly recheck it. Also show autophagy flux analysis in the presence of autophagy inhibitor Baf-A1.
A9. I have rechecked the tumor samples. Then, I have noticed that I mistook the resveratrol-treated sample for the non-treated sample. I am very sorry and I have revised them.
And I would also like to do autophagy flux analysis in the presence of autophagy inhibitor Baf-A1, but I have no time. Because I have to resubmit this revision by 10th November 2022. Then, I would essentially like to do that next time.
Q10. Quantify all western blots that make readability and analysis easier.
A10. I have quantified all Western blots.
Reviewer 3 Report
In their manuscript, the authors studied the effects of resveratrol, a polyphenolic antioxidant, in Oral cancers. The authors found that resveratrol induces autophagy in oral cancers by inhibiting SREBP1-mediated cell survival signaling. In addition, the authors showed that resveratrol suppresses the growth of oral cancers in vivo, without any cytotoxic effects.
While the manuscript is very well written, the following comments need to be addressed to make this manuscript suitable for publication:
- A major limitation of this manuscript is that most of the experiments include only one cell line and very often the same cell line. Key results presented in this manuscript should be retested in at least one more cell line. The assessment of the effect in vivo could also be performed using another cell line or using an immunocompetent mouse model to validate the results.
- Fig. 2: bands on the westerns should be quantified and presented as % of loading control.
- Fig. 4: the quality of the westerns is not optimal and the blots need to be quantified as described above.
- Fig. 5: western blots need to be quantified as described above.
- Fig. 8 : Quantification of the staining should be performed.
Author Response
Thank you for your advices to my manuscript.
- A major limitation of this manuscript is that most of the experiments include only one cell line and very often the same cell line. Key results presented in this manuscript should be retested in at least one more cell line. The assessment of the effect in vivo could also be performed using another cell line or using an immunocompetent mouse model to validate the results.
- I would like to retest using one more cell line and also to do the assessment of the effect in vivo using another cell line, but I have no time. Because I have to resubmit this revision by 10th November 2022. Then, I would essentially like to do that next time.
I have quantified all Western blots and added the densitometric analysis graphs in this manuscript.
I have also performed the quantification of staining and added the densitometric analysis graphs.
Round 2
Reviewer 3 Report
In their manuscript, the authors studied the effects of resveratrol, a polyphenolic antioxidant, in Oral cancers. The authors found that resveratrol induces autophagy in oral cancers by inhibiting SREBP1-mediated cell survival signaling. In addition, the authors showed that resveratrol suppresses the growth of oral cancers in vivo, without any cytotoxic effects.
While the manuscript has been revised, there are still some issues that need to be addressed to make this manuscript suitable for publication:
Major comments:
Figure 1: the authors should introduce, in the result section, the reason why they looked at SREBP1 and E-FABP in one or 2 sentences to give a bit of background to non-expert readers.
Since the authors used only one cell line throughout the entire paper (in vitro and in vivo), they should state in the discussion that their work is preliminary and suggest that in oral cancers, resveratrol has X and Y activities but that these results need to be confirm using more cell lines as well as other animal models.
Author Response
Major comments:
Q1. Figure 1: the authors should introduce, in the result section, the reason why they looked at SREBP1 and E-FABP in one or 2 sentences to give a bit of background to non-expert readers.
A1. In Figure 1 of the result section, I have changed to “E-FABP acts as a transporter of FAs which are required for cancer cell proliferation to provide new phospholipids for cytomembranes and that SREBP is a master regulator of genes involved in regulating FA synthesis and uptake. Then, differences in E-FABP and SREBP1 gene expression levels in HOSCC cell lines (HSC-2, HSC-3, HSC-4, Ca9-22, and SAS) were firstly analyzed by qRT-PCR. The results showed that Ca9-22 cells had the highest levels of E-FABP and SREBP1 mRNA expression among the HOSCC cell lines (Figure 1).”
Q2. Since the authors used only one cell line throughout the entire paper (in vitro and in vivo), they should state in the discussion that their work is preliminary and suggest that in oral cancers, resveratrol has X and Y activities but that these results need to be confirm using more cell lines as well as other animal models.
A2. In the last sentence of Discussion, I have added “This study is preliminary and suggests that in oral cancers, resveratrol has suppressive effects of lipid metabolism and anti-cancer activities but that these results need to be confirm using more cell lines as well as other animal models.”